# A Non-Uniform Interference-Fit Size Investigation of CFRP/Al Alloys by Riveting Mold Design

**Xingxing Wang** [1,2,*]**, Zhenchao Qi** [2]**, Mu Lu** [3] **and Haicheng Pan** [1]

1   College of Mechanical and Electrical Engineering, Suqian University, Suqian 223800, China
2   College of Mechanical and Electrical Engineering, Nanjing University of Aeronautics and Astronautics, Nanjing 210016, China
3   Jiangsu Geri Technology Co., Ltd., Nanjing 210036, China
*   Correspondence: wangxx@squ.edu.cn; Tel.: +86-18805199328

**Abstract:** The interference-fit size has a significant effect on the riveted lap joints of CFRP/Al alloy laminates. The requirements for the interference-fit size are different because of the strengthening of heterogeneous materials. However, in the riveting process of CFRP/Al alloys, the heterogeneous laminates lead to poor structural strength because of the different interference-fit size requirements. Therefore, differently assembled riveting molds are designed to acquire a novel interference-fit size, and the tensile test is adopted to evaluate their tensile properties. In addition, the fracture failure of CFRP/Al alloy laminate riveted lap joints is observed with an ultra-depth-of-field microscope. Finally, the best assembly type is identified as the trapezoid riveting mold combined with an arc riveting die, and the sidewall intersection angle of the trapezoid riveting mold is 66°, which could achieve a suitable interference-fit size and a better mechanical performance.

**Keywords:** CFRP/Al alloy; riveting mold assembly types; interference-fit size; mechanical performance

## 1. Introduction

Due to their high specific strength and modulus characteristics, carbon fiber-reinforced plastics (CFRPs) have been wildly applied in the transportation industry [1]. At present, CFRP/Al alloy riveted lap joints are used with an interference-fit size of 2~3% [2], which validly improves the connection performance of Al alloy sheets [3]. Mirzajanzadeh et al. [4] found that large interference-fit size can increase the fatigue life of Al alloy sheets in terms of its fretting fatigue crack properties. Abazadeh et al. [5] investigated bolted joints of Al alloy sheets, finding that a larger interference-fit size is beneficial to improving the performance of Al alloy sheets bolted joints.

However, a larger inference-fit size for CFRP/Al alloy riveted lap joints is a double-edged sword, because the interference-fit size of CFRP sheet riveted lap joints is typically less than 1.6% [6,7]. A large interference-fit size will induce extrusion, instability, and delamination of the hole surface of CFRPs, particularly when the entrance of the connection surface is over 2% [8,9]. Khashaba et al. [10] proposed a model to predict the 3D progressive damage of the clearance-fit sizes and static strength of CFRP joints. Chen et al. [11] studied the effects of zero-fit, clearance-fit, and interference-fit sizes on the mechanical performance of CFRP joints. The results showed that the joints with zero-fit or clearance-fit had a better shear performance than those with the interference-fit. Zou et al. [12] developed a FEM model to predict the effect of interference-fit size on delamination defects. The results showed that increases in the interference-fit size worsened the delamination of CFRP.

The abovementioned studies examine homogeneous laminates, hence the interference-fit size could be simply adjusted to suit them. Nevertheless, during the riveting of heterogeneous laminates of CFRP/Al alloys, their interference-fit size is different from that of homogeneous laminates [13]. The traditional riveting process is still used, which achieves an interference-fit size that is too large for CFRPs but too small for Al alloys. Therefore,

researchers have studied ways to improve the mechanical performance of riveted lap joints in CFRP/Al-alloy laminates. Cui et al. [14] investigated the effect of a trapezoidal riveting mold on interference-fit size; the results proved that mold angle had a significant effect on interference-fit size. Jiang et al. [15] studied CFRP/Al-alloy laminates by electromagnetic riveting with different riveting molds; the results showed that the 80° trapezoidal riveting mold can not only improve interference-fit size but also the fatigue performance. Ma et al. [16] developed an effective way to avoid joint cracking by optimization of the riveting mold structure. Although scholars have adopted the riveting mold design to investigate the interference-fit size and achieve better mechanical performance, the interference-fit sizes of CFRP sheets and Al alloy sheets are still the same without considering the difference in material. How to make the interference-fit size of CFRP/Al alloys more suitable still needs further research. Therefore, the optimal die design can not only improve material flow and mechanical properties but can also reduce the forming load [17–19]. In addition, to shorten the cycle and reduce costs, many scholars have adopted the FEM and experimental methods to carry out research [20–22].

It has been acknowledged that increases in the interference fit of CFRP riveted lap joints leads to fiber instability and weakens the performance [23,24]. Therefore, this paper not only investigates the design parameters of the riveting mold but also the assembly types of the riveting mold. Primarily, the design parameters of the riveting mold are analyzed by FEM, and the significant factor of process parameters are confirmed. Furthermore, riveting experiments are carried out and the interference-fit size is measured. Finally, the tensile test is performed to determine the structure of the riveting mold, and the fatigue failure types and microstructure performance are observed.

## 2. Materials and Methods

### 2.1. Sample Preparation

The T700 CFRP and 2024 Al alloy were selected for the adapting piece; the material of the rivet is Ti-45Nb. T700 CFRP was used as a unidirectional carbon fiber/epoxy with a thickness of 0.15 mm per ply (provided by GW COMPOS Company Ltd., Weihai, China). The thickness of fabricated T700 CFRP laminate is 2.3 mm with 16 piles, the ply orientation of T700 CFRP is [0°/90°/45°/−45°/−45°/45°/90°/0°] 2 s, and the weight fraction of carbon fiber is about 60%. The material properties of the fabricated CFRP laminates are presented in Table 1. Moreover, the fabricated Ti-45Nb rivets (provided by CAG Company Ltd., Beijing, China) were annealed by heating in a vacuum (less than 0.1 um mercury) to a temperature within the range of 1450 °F to 1600 °F, and held at heat for sufficient time to produce a recrystallized structure that will meet the requirements of Ti-45Nb's properties. The material properties of the Ti-45Nb rivets are presented in Table 1. In addition, the diameter of the Ti-45Nb rivet is 4 mm, and the prefabricated hole diameters of the CFRP laminates were drilled using a dagger drill with a diameter of 4.1 mm, and the aperture of the sample was measured using a plug gauge. The sizes of CFRP riveted specimens according to the ASTM D5661 are shown in Figure 1, and W/D ≥ 6, E/D ≥ 3.

**Table 1.** Mechanical properties of the sample.

| CFRP Laminates | | Ti-45Nb Rivets | |
|---|---|---|---|
| **Property** | **Value** | **Property** | **Value** |
| Resin content (%) | 40 | Density [g/cm$^3$] | 5.7 |
| Tensile strength (MPa) | 2300 | Poisson ratio | 0.34 |
| Tensile modulus (GPa) | 115 | Tensile modulus [GPa] | 62 |
| Flexural strength (MPa) | 1250 | Yield strength [MPa] | 425 |
| Compressive strength (MPa) | 1050 | Tensile strength [MPa] | 570 |
| Interlaminar shear strength (MPa) | 55 | | |

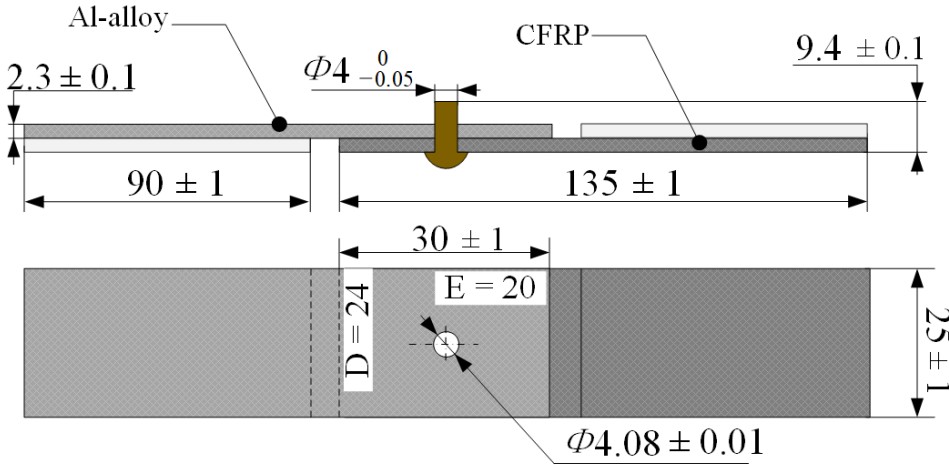

**Figure 1.** Dimensions of the riveted lap joints of CFRP laminates (dimensions in mm).

*2.2. Handling Method for the Experimental Results*

It has been acknowledged that large interference-fit sizes induce CFRP damage; hence, the ideal interference-fit size for CFRP/Al-alloy laminates is as shown in Figure 2, where the suitable interference-fit size of area 1 is 2.0% to 2.5%, the suitable interference-fit size of area 2 is about 1.8%, and area 3 adopts a zero fit or clearance fit. The interference-fit size of CFRP should be less than that of Al alloys. Therefore, to acquire the ideal fit for CFRP/Al-alloy riveted lap joints, the mold of the rivet mechanical head (bottom mold) is used as an arc riveting mold (ARM) to prevent the deformation of rivet mechanical head, resulting in a clearance fit at the entrance of the CFRP connection surface. The mold of the rivet bar (top mold) is used for a flat riveting mold (FRM) or a trapezoid riveting mold (TRM); hence, the assembly types of the riveting mold are shown in Figure 3. During the riveting process, the interface slip of the die and rivet would produce a radial constraint force, which can promote filling of the hole by a material. Considering the larger interference-fit size of Al alloys, riveting molds of type-2 assembly were adopted to research the variation in interference-fit size.

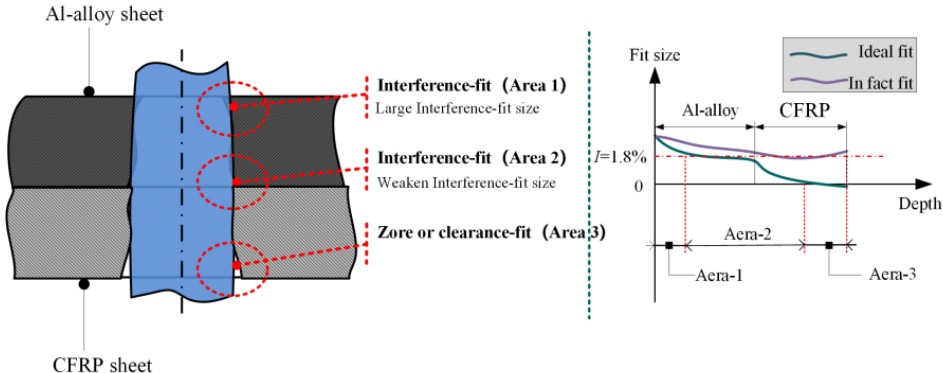

**Figure 2.** Two interference-fit curves of composite/Al alloy sheets.

After the riveting, the deformed rivet specimens are cut with diamond blades. The interference-fit size of the deformed rivet bar was measured by a Vernier caliper with an accuracy of 0.01 mm. Subsequently, the DBSL-10t tensile test machine was used to test the mechanical properties of the riveted specimens under pull-out loading based on the ASTM-5961 standard. Finally, the fracture morphology of the microstructure was observed by the RH-2000 super-depth microscopy system (Haoshi Instrument Technology Co., Ltd., Shanghai China).

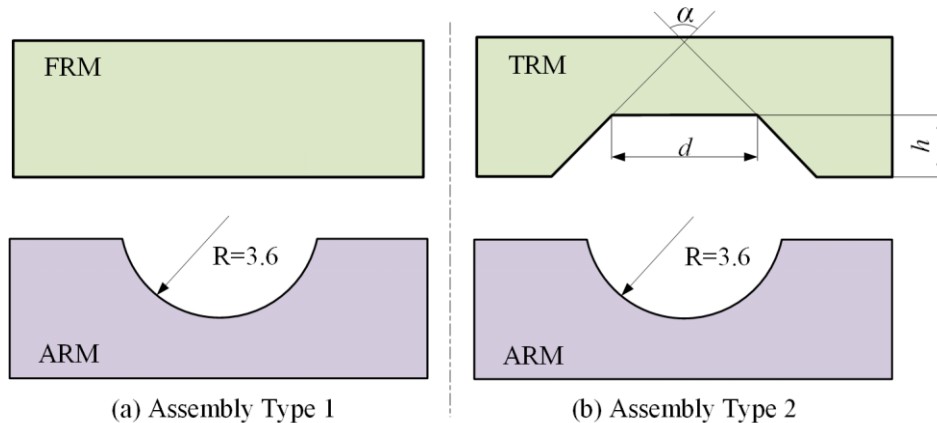

(a) Assembly Type 1      (b) Assembly Type 2

**Figure 3.** Different riveting mold assembly types: (**a**) FRM-ARM, (**b**) TRM-ARM.

## 3. Results

### 3.1. River Die Combination Types

The combination-type design of the rivet die is an effective way to achieve a non-uniform interference fit in CFRP/Al alloy sheets. As shown in Figure 4a, the effect of rivet die combination types on the tendency of interference sizes was preliminarily analyzed using FEM, which included four kinds of types, i.e., FRM to FRM, FRM to ARM, TRM to FRM, and TRM to ARM. It could be seen that the TRM-ARM or FRM-ARM types could obtain a non-uniform interference fit. Meanwhile, to ensure that the non-uniform interference sizes are reasonable, the TRM-ARM type needs to be optimized. Figure 4b shows the structure of the TRM-ARM type. The parameters of the trapezoid rivet dies include the sidewall intersection angle ($\alpha$), sidewall height ($h$), and upper diameter of the sidewall ($d$). According to the constant volume principle, the structural parameters of TRMs have significant influences on the material filling into the pre-drilled hole. The arc rivet die matched with the rivet manufactured head, where the radius $R_0$ and the depth $h_0$ are 3.6 mm 2.0 mm, respectively.

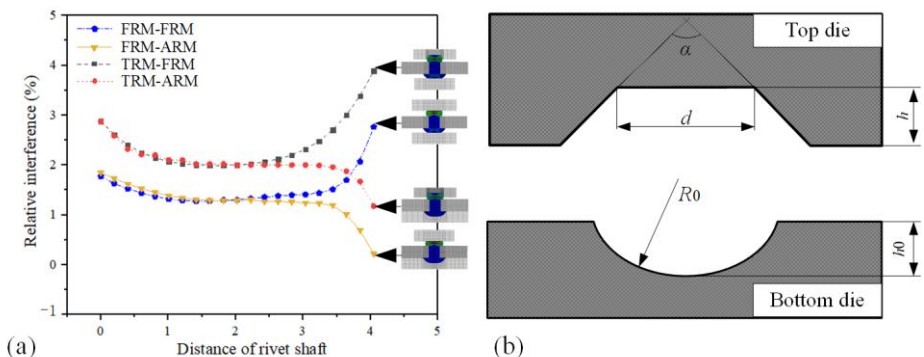

**Figure 4.** Simulation result of different riveting mold assembly types (**a**) Interference-fit tendency for different combinations of rivet dies. (**b**) The rivet die assembly types for a non-uniform interference-fit size.

### 3.2. Rivet Radial Force Constrain Modeling

In the riveting process, the interface slip of the die and rivet would produce a radial constraint force which can promote material filling the hole. Considering the isotropic material and axisymmetric structure of the rivet, the radial force presents a homogeneous distribution. The diagram of the central section radial force was presented in Figure 5. The rivet shaft is subject to the axial force ($F_z$) and radial force ($F_r$), and the $F_r$ is $f_u$ in a flat rivet die, which can improve the interference size. The $F_z$ is a constant; hence, adopting a concave rivet die structure to increase $F_r$ is an effective method.

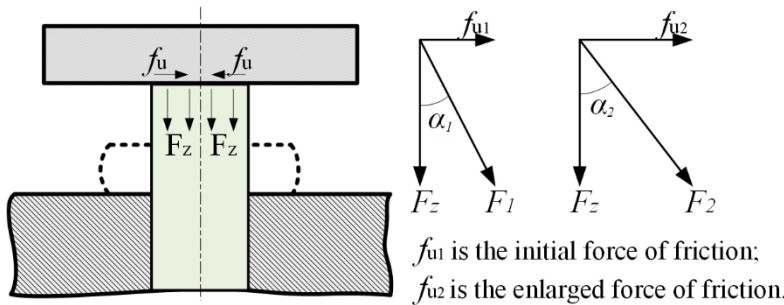

Figure 5. A diagram of the radial force in the central section.

In Figure 6, the comparison of radial constraints force between FRM and TRM will be analyzed as follows. As shown in Figure 6a, the radial constrain force of a flat rivet die ($F_{frm}$) is simple friction ($f_u$), as following Equation (1):

$$F_{frm} = f_u = \frac{1}{2} F_z \times m \tag{1}$$

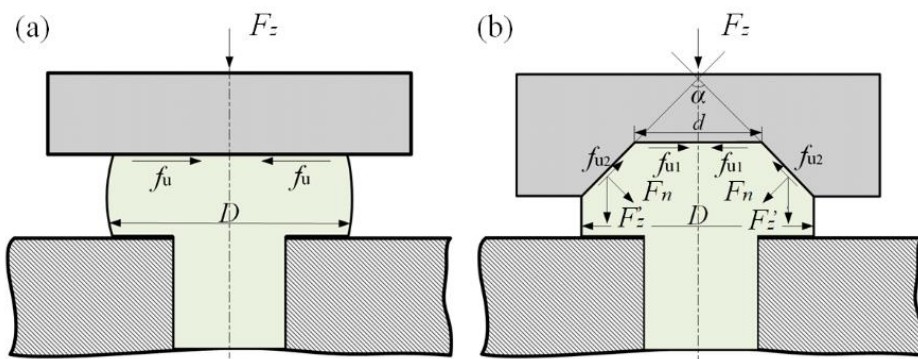

Figure 6. Comparison analysis of radial force. (a) Flat rivet mold. (b) Special rivet mold.

As shown in Figure 6b, the radial constrain force of trapezoid rivet to die ($F_{trm}$) includes horizontal friction ($f_{u1}$), the horizontal component of $f_{u2}$ and $F_n$, as following Equation (2):

$$\begin{cases} F_{trm} = F_r = f_{u1} + f_{u2} \times \sin\frac{a}{2} + F_n \times \cos\frac{a}{2} \\ f_{u1} = \frac{1}{2} F_z \times \frac{d}{D} \times m \\ f_{u2} = F_z' \times \cos\frac{a}{2} + F_n \times m \\ F_n = F_z' \times \sin\frac{a}{2} \\ F_z' = \frac{1}{2} F_n \times \frac{D-d}{D} \end{cases} \tag{2}$$

Therefore, when the $F_{trm}$ is larger than $F_{frm}$, it is possible to combine Equation (1) with Equation (2) to obtain the function of $\alpha$ and $\mu$, as following Equation (3):

$$2\tan\frac{\alpha}{2} > \mu \tag{3}$$

where $0 < \alpha < 180°$, and $\mu \approx 0.2$ (normally, the friction coefficient of upsetting is 0.2), then we substituted them into Equation (3) and obtained $12° < \alpha < 180°$.

## 4. Discussion

In Figure 4, the radius $R_0$ of the ARM is 3.6 mm based on the size of the rivet mechanical head. However, the parameters of TRM include the sidewall intersection angle ($\alpha$), sidewall height ($h$), and upper diameter of sidewall ($d$), the level of the parameters is shown in

Table 2, and total schemes are 16 ($L3^4$). The FEM model is established by Deform-3D, as shown in Figure 7a, and the variation of interference-fit sizes is shown in Figure 7b–f. The maximum interference-fit size (Imax) is counted in Table 3.

**Table 2.** Parameters level of the TRM structure.

| Variable | Level 1 | Level 2 | Level 3 | Level 4 |
|---|---|---|---|---|
| $h$/mm | 1.6 | 1.8 | 2.0 | 2.2 |
| $d$/mm | 4.2 | 4.4 | 4.6 | 4.8 |
| $\alpha$ | 22° | 44° | 66° | 88° |

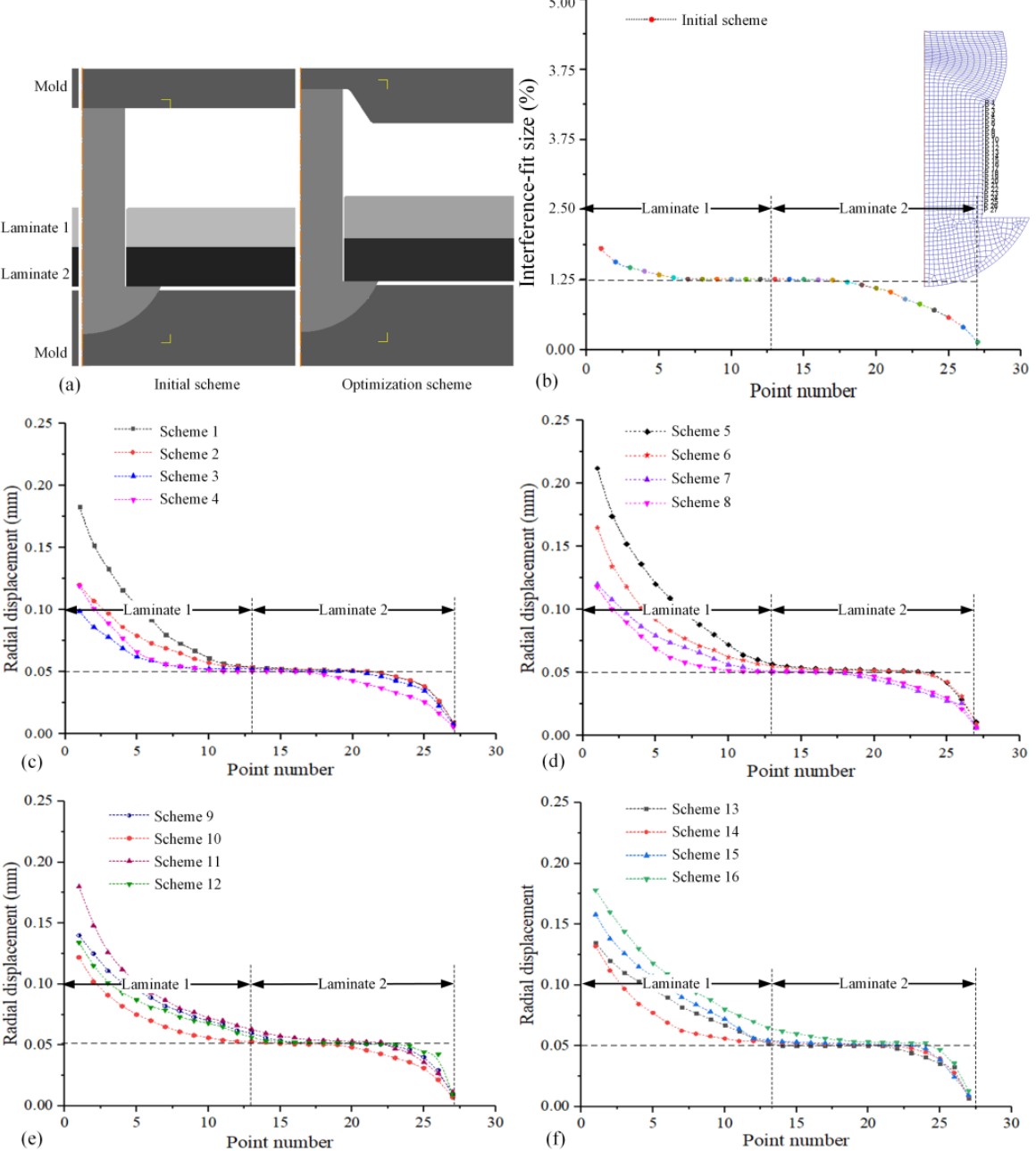

**Figure 7.** The variation process of the interference-fit size: (**a**) FEM model; (**b**) initial scheme; (**c**) schemes 1–4; (**d**) schemes 5–8; (**e**) schemes 9–12; (**f**) schemes 13–16.

**Table 3.** The schemes and results of the research design.

| Scheme | $h$/mm | $d$/mm | $\alpha$/° | $I_{max}$/% |
|--------|--------|--------|-----------|-------------|
| 1 | 1.6 | 4.2 | 22 | 3.70 |
| 2 | 1.6 | 4.4 | 44 | 2.49 |
| 3 | 1.6 | 4.6 | 66 | 2.01 |
| 4 | 1.6 | 4.8 | 88 | 2.45 |
| 5 | 1.8 | 4.2 | 44 | 4.37 |
| 6 | 1.8 | 4.4 | 22 | 3.41 |
| 7 | 1.8 | 4.6 | 88 | 2.34 |
| 8 | 1.8 | 4.8 | 66 | 2.30 |
| 9 | 2.0 | 4.2 | 66 | 2.92 |
| 10 | 2.0 | 4.4 | 88 | 2.48 |
| 11 | 2.0 | 4.6 | 22 | 3.65 |
| 12 | 2.0 | 4.8 | 44 | 2.81 |
| 13 | 2.2 | 4.2 | 88 | 2.72 |
| 14 | 2.2 | 4.4 | 66 | 2.70 |
| 15 | 2.2 | 4.6 | 44 | 3.15 |
| 16 | 2.2 | 4.8 | 22 | 3.58 |

The deviations of the parameters are listed in Table 4. It could be seen that the parameters have a significant influence on interference-fit size in the order of $\alpha > d > h$. In Figure 7, the collecting points of the rivet bar are shown in the red circle. Comparing the FRM with the TRM, the TRM significantly improves the interference-fit size, and the interference-fit size in the entrance of the CFRP connection surface is reduced by the ARM. Then, the decision tree model is adopted to train the data. The trained result of weight for variables are displayed in Figure 8. It could be seen that the average weight values of $\alpha$, $d$, and $h$ for the interference-fit size are 0.65, 0.24, 0.11, respectively. In summary, the weight values of $\alpha$ for load and interference are the most significant.

**Table 4.** The $I_{max}$ average of each variable.

| Level | $h$/mm | $d$/mm | $\alpha$ |
|-------|--------|--------|----------|
| 1 | 2.662 | 3.428 | 3.585 |
| 2 | 3.105 | 2.770 | 3.205 |
| 3 | 2.965 | 2.788 | 2.482 |
| 4 | 3.038 | 2.785 | 2.498 |
| Deviation max-min | 0.443 | 0.657 | 1.103 |

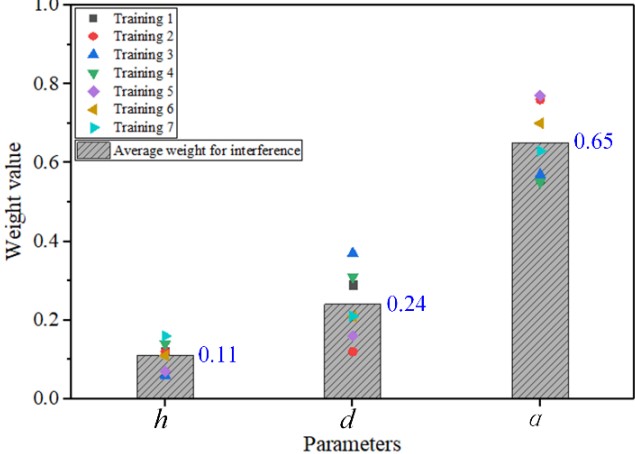

**Figure 8.** Weight of interference-fit size determined by decision-making tree.

According to Figure 7c–f, scheme 2, scheme 6, scheme 9, and scheme 15 are reasonable in terms of interference fit-size. Considering the effect of parameter *a*, the schemes 6, schemes 9, and schemes 15 are adopted to implement the experimental verification.

### 4.1. Non-Uniform Interference-Fit Size

According to the interference size effect analysis, the TRM based on the parameters of scheme 6, scheme 9, and scheme 15 are manufactured, and the manufactured FRM is used as the contrast experiment, as shown in Figure 9. The force and speed of the riveting process are 14.5 kN and 10 mm/s, respectively. The interference-fit size of each specimen is measured at five positions, as shown in Figure 10. The measured interference-fit sizes, with three repetitions, are listed in Table 5, and the average interference-fit size ($I_A$) was calculated.

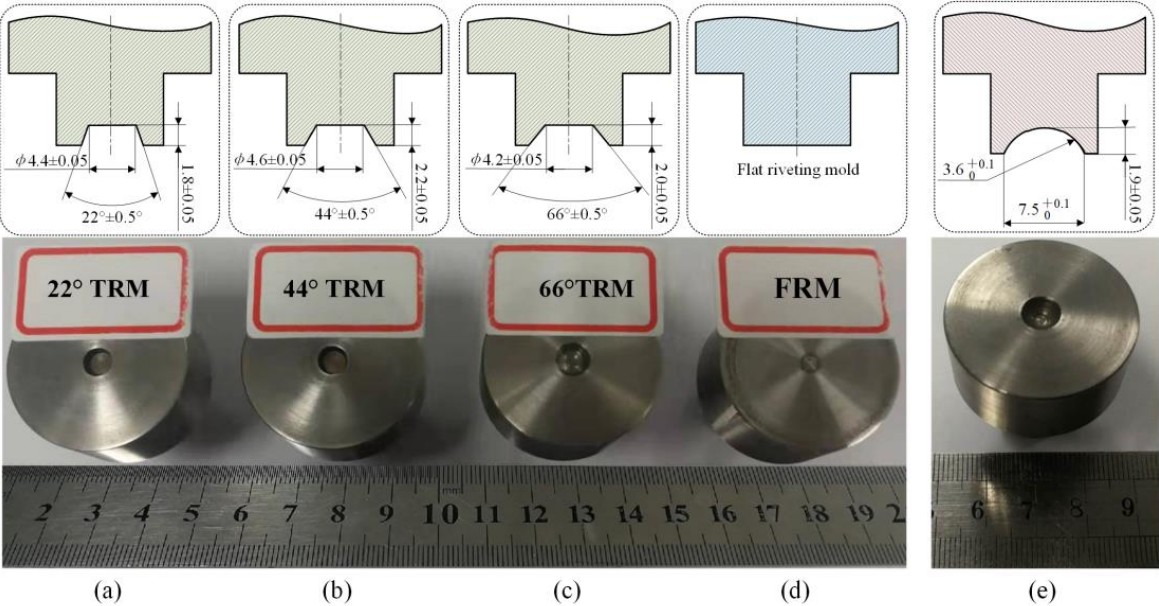

**Figure 9.** Dimensions and products of different riveting molds: (**a**) 22° TRM, (**b**) 44° TRM, (**c**) 66° TRM, (**d**) FRM, (**e**) ARM.

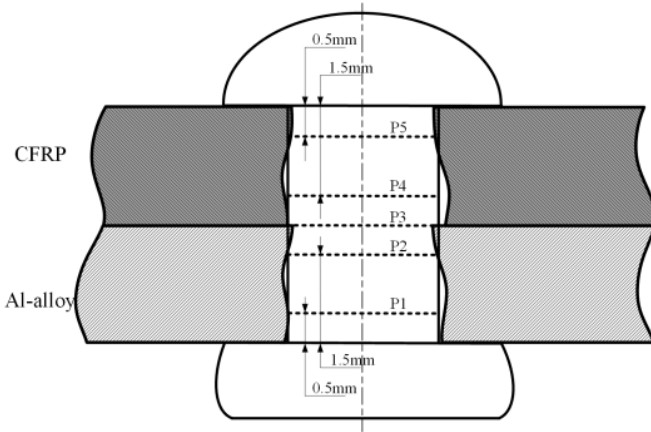

**Figure 10.** Measurement positions for Interference-fit size.

**Table 5.** The measured results of relative interference for different rivet dies.

| Type | Position | Repeat 1 (mm) | Repeat 2 (mm) | Repeat 3 (mm) | Average Value (mm) | $I_A$ (%) |
|---|---|---|---|---|---|---|
| FRM | 1 | 4.16 | 4.17 | 4.16 | 4.163 | 2.03 |
| | 2 | 4.13 | 4.13 | 4.13 | 4.130 | 0.98 |
| | 3 | 4.12 | 4.10 | 4.12 | 4.130 | 0.98 |
| | 4 | 4.11 | 4.12 | 4.11 | 4.113 | 0.809 |
| | 5 | 4.10 | 4.12 | 4.12 | 4.113 | 0.809 |
| 22° TRM | 1 | 4.19 | 4.20 | 4.18 | 4.190 | 2.70 |
| | 2 | 4.17 | 4.16 | 4.16 | 4.163 | 2.03 |
| | 3 | 4.14 | 4.15 | 4.15 | 4.147 | 1.64 |
| | 4 | 4.13 | 4.12 | 4.13 | 4.127 | 1.15 |
| | 5 | 4.10 | 4.10 | 4.10 | 4.100 | 0.74 |
| 44° TRM | 1 | 4.21 | 4.20 | 4.21 | 4.207 | 3.11 |
| | 2 | 4.18 | 4.17 | 4.19 | 4.180 | 2.45 |
| | 3 | 4.15 | 4.15 | 4.15 | 4.150 | 1.72 |
| | 4 | 4.14 | 4.13 | 4.14 | 4.137 | 1.40 |
| | 5 | 4.12 | 4.13 | 4.13 | 4.127 | 1.15 |
| 66° TRM | 1 | 4.19 | 4.18 | 4.19 | 4.187 | 2.63 |
| | 2 | 4.18 | 4.18 | 4.17 | 4.177 | 2.38 |
| | 3 | 4.14 | 4.14 | 4.13 | 4.137 | 1.40 |
| | 4 | 4.14 | 4.13 | 4.14 | 4.137 | 1.40 |
| | 5 | 4.13 | 4.11 | 4.13 | 4.123 | 1.05 |

To intuitively analyze the data in Table 5, the date histograms of different riveting molds are implemented in Figure 11. Figure 11a displays the interference-fit size of FRM-ARM, which shows that the maximum interference-fit size is lower than 2%, and the CFRP sheet fit well. However, the interference-fit size of the Al alloy sheet does not meet the requirements. In Figure 11b,c, the results show that the variation tendency of interference-fit size for the 22° TAM-ARM and 44° TAM-ARM had good consistency. However, both independently resulted in non-uniform interference-fit sizes in Al-alloy and CFRP. In Figure 11d, the variation tendency of interference-fit size for the 66° TAM-ARM presents good consistency with the ideal interference-fit size. In addition, the interference-fit size is relatively uniform for each laminate. Therefore, based on the variation tendency of interference-fit size, the 66° TAM-ARM is a better assembly type compared to the others, which is consistent with the FEM result.

### 4.2. Strength and Fracture Modes

To research the effect of interference-fit size with different riveting molds on the strength of CFRP/Al-alloy riveted lap joints, the tensile test is carried out at a speed of 5 mm/min. The tensile load-displacement curves of 22° TRM, 44° TRM, and 66° TRM are displayed in Figure 12. It could be seen that the tensile load rises rapidly with the increase in displacement in the elastic deformation stage. However, in the failure stage of the load-displacement curves there were distinct differences, especially considering the specimen riveted by 66° TRM. For the specimens riveted by 22°-TRM and 44° TRM, the pulled-off failure displacement is longer than that of 66° TRM. In addition, the maximum tensile load (5734 N) of the specimen is riveted by 66° TRM, which is a little higher than that of 44° TRM (5709 N); both of them are larger than the maximum tensile load (5118 N) of 22° TRM.

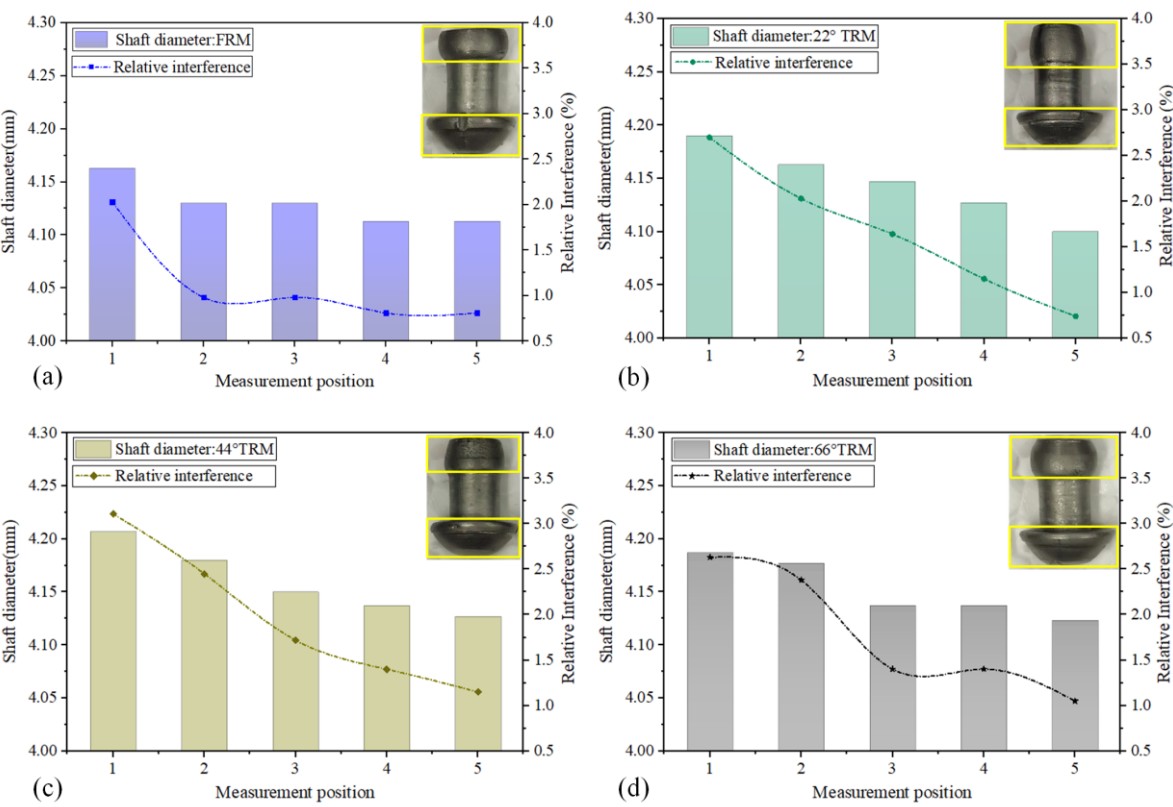

**Figure 11.** Diameter and interference-fit size results with different riveting molds: (**a**) FRM, (**b**) 22° TRM, (**c**) 44° TRM, (**d**) 66° TRM.

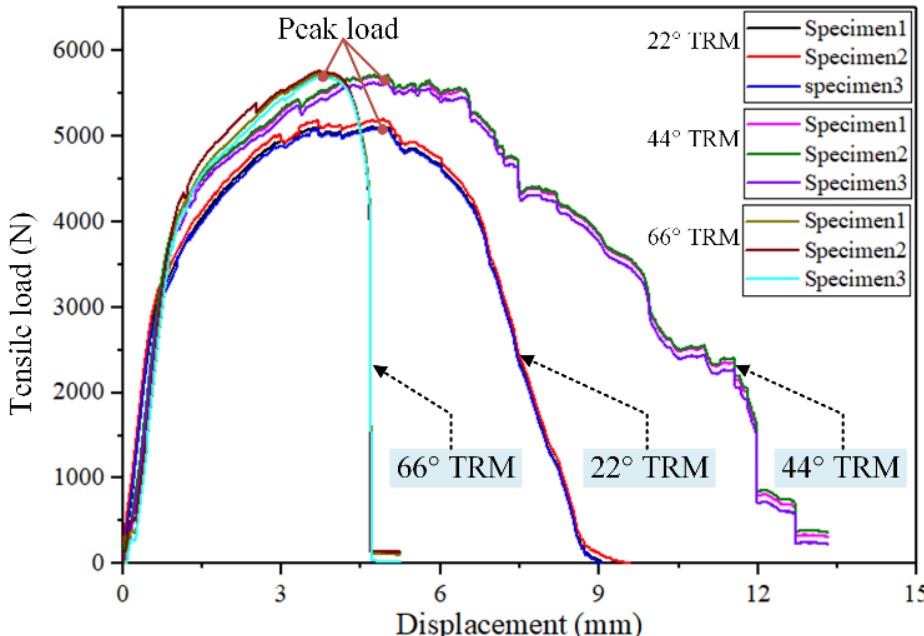

**Figure 12.** The tensile load-displacement curves with different TRM angles.

Furthermore, the failure types of specimens with different TRMs are shown in Figure 13. It could be seen that the failure type of specimens with the 22° TRM and 44° TRM is rivet pull-out, but the failure type of specimens with the 66°-TRM is rivet shear fracture. This induces a longer failure displacement in the tensile test for specimens with the 22° TRM and 44° TRM, and the failure displacement for specimens with the 66° TRM is short. Combined with Figure 11 to reveal the difference in failure type, for specimens with the

22° TRM and 44° TRM, the interference-fit size at the entrance of Al alloy in the 44° RM specimen is larger than that of the 22° TRM under the same riveting force. Hence, the strength of the Al-alloy riveted lap joint with the 44° TRM is larger than that of the 22° TRM, and the strength of the Al-alloy riveted lap joint is larger than the CFRP riveted lap joint with 44° TRM, and the strength of the Al alloy and CFRP with the 22° TRM follows the opposite pattern. In addition, the interference-fit size is non-uniform for each sheet. Both reasons induce rivet pull-out from the sheet. However, the relative uniform interference-fit size provided by the 66° TRM effectively reinforces the fit strength of the Al alloy and CFRP and the interference-fit size requirement is satisfied, causing the rivet to shear fracture.

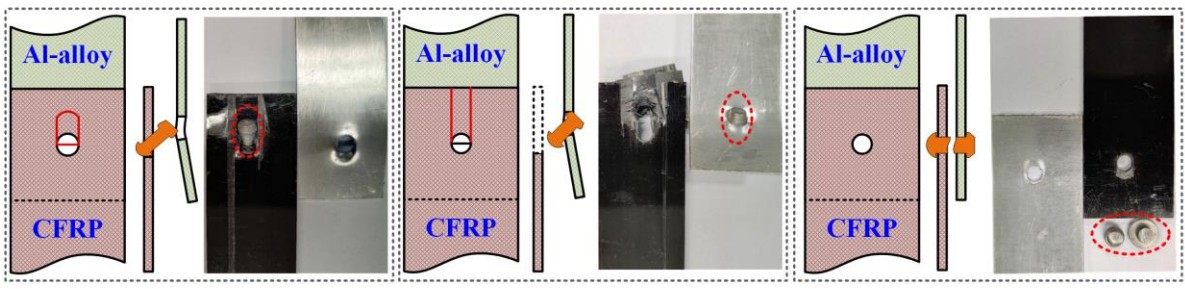

(a) Rivet pull-out from Al-ally      (b) Rivet pull-out from CFRP      (c) Rivet shear fracture

**Figure 13.** The typical fracture modes for different riveting molds: (**a**) FRM and 22° TRM, (**b**) 44° TRM, (**c**) 66° TRM.

*4.3. Fracture Microstructure*

The failure morphology was observed using a microscope. In Figure 14, the microstructure of the specimens' pull-out hole by the 22° TRM is observed. The hole of the Al alloy was stretched, and the hole appears to be relatively smooth and does not appear to crack, remaining in the plastic deformation extension stage. The positions of the observed CFRP are shown in Figure 14a, corresponding to Figure 14d–g, respectively. It could be seen that the carbon fiber at the $P_1$ position did not sustain delamination or extrusion damage. The $P_2$ position shows that the $45°/−45°/90°$ carbon fibers suffered from extrusion. Carbon fibers at the $P_3$ position were subjected to tension and extrusion, which caused a part of the carbon fibers to snap. At the $P_4$ position, the CFRP hole sustained severe extrusion, and delamination and carbon fiber breakage appeared.

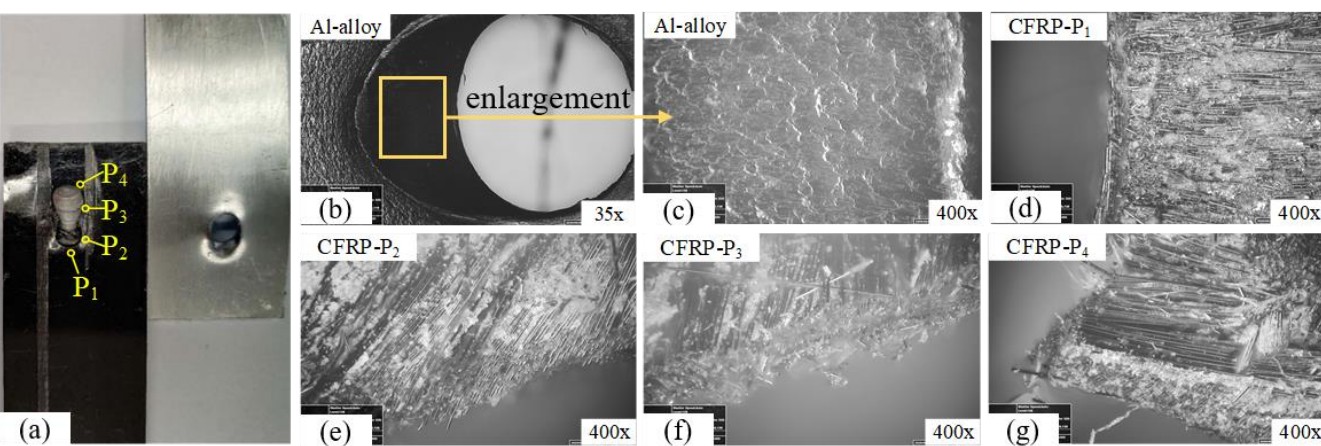

**Figure 14.** The specimen fracture morphology of the 22° TRM: (**a**) Observation diagram; (**b**) the hole morphology of Al-alloy; (**c**) the enlarged hole morphology of Al-alloy; (**d**) CFRP $P_1$ area; (**e**) CFRP $P_2$ area; (**f**) CFRP $P_3$ area; (**g**) CFRP $P_4$ area.

The failure microtopography of specimens with the 44°-TRM is shown in Figure 15. It could be seen that the CFRP's damage is much more serious than that of the specimens

with the 22° TRM; the observed positions are shown in Figure 15a. In Figure 15b, it could be seen that the carbon fibers in $P_1$ have been peeled off. As the rivet was pulled out, the carbon fibers were sheared, resulting in severe delamination defect in the CFRP, as shown in Figure 15c,d. In Figure 15e, the $P_4$ position of the CFRP sheet sustained severe fracture, where carbon fibers were seriously snapped and crushed.

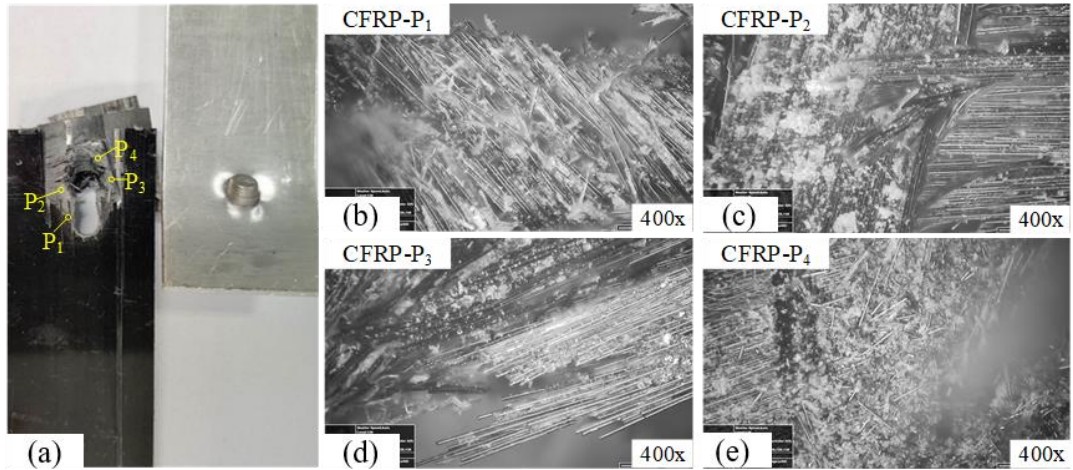

**Figure 15.** The specimen fracture morphology of the 44° TRM: (**a**) Observation diagram; (**b**) CFRP $P_1$ area; (**c**) CFRP $P_2$ area; (**d**) CFRP $P_3$ area; (**e**) CFRP $P_4$ area.

The failure mode of the specimen with the 66° TRM is shown in Figure 16. It could be seen that the CFRP and Al-alloy remain intact, and the rivet sustains shear fracture. In Figure 16a, according to the symmetrical fracture surface of the rivet, it is divided into three zones, i.e., the shear source (Zone 1), ductile fracture (Zone 2), and brittle fracture (Zone 3). In Figure 16b, the fracture microstructure in Zone 1 is relatively smooth and has a distinct transition area. The transition area displays an elongating shear-long micro-pit, then gradually develops into a ductile fracture area and brittle fracture area, as shown in Figure 16c. The ductile fracture in Zone 2 is a shear-long micro-pit, as shown in Figure 16d. It indicates that the material has undergone a severe shear deformation under a low strain ratio. The brittle fracture morphology in Zone 3 is shown in Figure 16e. It is a typical intercrystalline delamination fracture, which indicates that the Ti-45Nb rivet has poor plasticity.

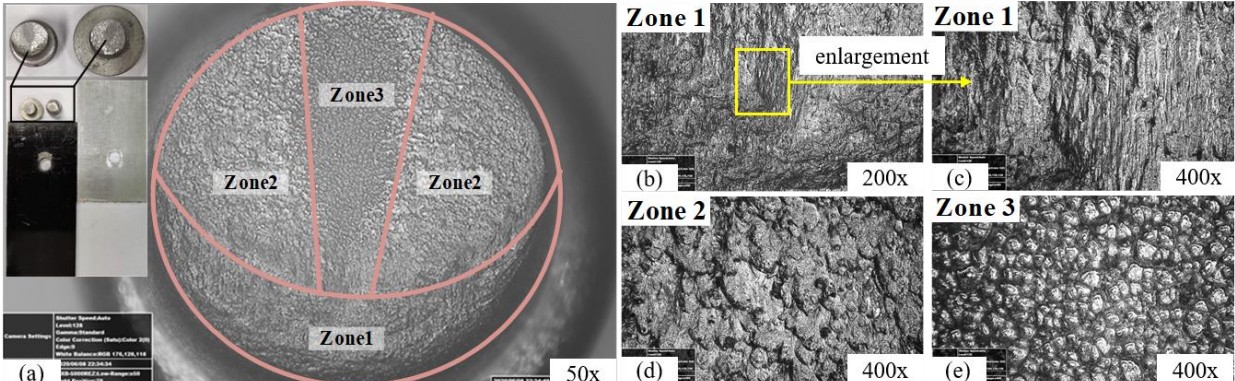

**Figure 16.** The fracture morphology the specimen with 66° TRM. (**a**) Observation diagram; (**b**) Zone 1 microtopography of Ti-alloy rivet; (**c**) the enlarged Zone 1 microtopography of Ti-alloy rivet; (**d**) Zone 2 microtopography of Ti-alloy rivet; (**e**) Zone 3 microtopography of Ti-alloy rivet.

## 5. Conclusions

In this research, rivet mold assembly types are used to investigate the effect of interference-fit size on the mechanical performance of CFRP/Al-alloy riveted lap joints. The main conclusions are as follows:

(1) The FEM results show that the TRM-ARM assembly type can achieve an ideal fit for the CFRP/Al-alloy riveted lap joint; the TRM design parameter with the most significant effect on interference-fit size is the sidewall intersection angle ($\alpha$); the average weight value of $\alpha$ for the interference-fit size is 0.65.

(2) The experimental results show that the TRM-ARM can acquire a larger interference-fit size in an Al alloy sheet compared to the FAM-ARM; the 66° TRM-ARM assembly type has a more uniform interference-fit size for each CFRP and Al-alloy laminate, and the fit surface of the hole is better reinforced relative to the 22° TRM-ARM and 44° TRM-ARM.

(3) The tensile tests show that the 66° TRM-ARM achieves a better shearing performance than the 22° TRM-ARM and 44° TRM-ARM.

**Author Contributions:** Methodology, X.W.; validation, Z.Q.; formal analysis, X.W.; investigation, X.W., M.L. and H.P.; data curation, X.W.; writing—original draft preparation, X.W., Z.Q. and H.P.; writing—review and editing, X.W.; funding acquisition, X.W. and H.P. All authors have read and agreed to the published version of the manuscript.

**Funding:** This work was supported by the Basic Science Research Project of Jiangsu Province (No. 22KJB460008); Suqian Sci&Tech program (Grant No. K202210); Suqian Sci&Tech program (Grant No. Z2021139).

**Data Availability Statement:** Not Appliable.

**Conflicts of Interest:** On behalf of all authors, the corresponding author states that there is no conflict of interest.

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
