# Peer review of "A Non-Uniform Interference-Fit Size Investigation of CFRP/Al Alloys by Riveting Mold Design"

_processes, doi:10.3390/pr11030962_

Round 1
Reviewer 1 Report
It’s my honor to review the article entitled " A non-uniform interference-fit size investigation of CFRP/Al-alloy by riveting mold design" (ID: 2274957). The different assembly types of riveting mold are designed to acquire a novel interference-fit size, and the tensile test is adopted to evaluate the tensile property. In addition, the fracture failure of CFRP/Al-alloy laminates riveted lap joint is observed by the ultra-depth of field microscope. Finally, the novel assembly type is the 66°TAM-ARM. This manuscript provides a new scheme for reliable riveting of CFRP/Al-alloy and other different materials.
So I recommend to you that this manuscript can be accepted for publication after some major revised efforts have been done. The following is the list of detailed comments with respect to the paper:
1. Some describes for many studies details need to be further improved. The authors should pay attention to English grammar, punctuation marks and capital and small letters. Such as “An non-uniform...... ” in title, “Mpa” and “cm3”in Table 1. L92, L108, L106, L105 and L106,L115-118, L124, “2.3 mm” and “10mm”, “14.5KN”……
2. It is necessary to improve the INTRODUCT and provide relevant references for the past three years.
3. Please explain “produce a recrystallized structure that will meet the requirements of 3.5” in Line 83?
4. What is the best depth proportion of the three areas in ideal fit? Whether the experiment results are consistent with the optimum ratio is discussed?
5. The expression of some special words should be unified in the text, figures and tables. Such as “trapezoid riveting mold (TRM)” and “trapezoid riveting die(TRD)” , “arc rivet die (ARD)” and “arc rivet mold(ARM)”in the text.
6. There are many statements in the text that are inconsistent with the figure. Please correct them. Such as L87 Fig.3, L115 (Fig. 3a), L144(Fig.7b), L180(scheme 6, scheme 9, and scheme 15) and Fig. 9……
7. All the references in this paper should be formatted according to journal specification.
Reviewer 2 Report
In this paper, a non-uniform interference-fit size for CFRP/Al-alloy laminates riveted lap joints was developed due to the different interference-fit size requirements for CFRP and Al-alloy. According to the FEM and experimental results, the best assembly type is the trapezoid riveting mold combined with an arc riveting die, and the sidewall intersection angle of the trapezoid riveting mold is 66° which could achieve a suitable interference-fit size and a better mechanical performance. Overall, the paper has clear thinking and is well written. It could be accepted for publication after some minor revisions. Some comments are listed below.
Introduction section:
1) “An non-uniform interference-fit size” should be “A non-uniform…”.
2) “Large interference-fit size can induce extrusion” should be “the large interference-fit size will induce extrusion”
3) “Therefore, researchers have studied the way to improve the mechanical performance of CFRP/Al-alloy laminates riveting lap joints” is not appropriate, it should describe the interference-fit size.
4) Some grammatical mistakes need to be modified for this section.
Experimental materials and methods
1) Why only display the mechanical properties of CFRP and rivet?
Results
1) Some grammatical mistakes need to be modified for this section, such as “constrain modelling”, “In riveting process”
Discussion
1) The subtitle is not suitable. Please check.
2) How to make sure the parameters level of TRM structure for simulation.
3) In table 4, what the meaning of 1103? Please check.
Round 2
Reviewer 1 Report
Some words in Fig.7 are not clear enough.